# Bio-Inspired and Smart Nanoparticles for Triple Negative Breast Cancer Microenvironment

**DOI:** 10.3390/pharmaceutics13020287

**Published:** 2021-02-22

**Authors:** Mahsa Keihan Shokooh, Fakhrossadat Emami, Jee-Heon Jeong, Simmyung Yook

**Affiliations:** 1Department of Pharmaceutics, College of Pharmacy, Tehran University of Medical Sciences, Tehran 1417614411, Iran; mkshokouh@gmail.com; 2College of Pharmacy, Keimyung University, Daegu 42601, Korea; f-emami@razi.tums.ac.ir; 3College of Pharmacy, Yeungnam University, Gyeongsan, Gyeongbuk 38541, Korea

**Keywords:** triple negative breast cancer, targeted therapy, smart nanoparticles, cell membrane coated nanoparticles, immune cell targeted nanoparticles

## Abstract

Triple negative breast cancer (TNBC) with poor prognosis and aggressive nature accounts for 10–20% of all invasive breast cancer (BC) cases and is detected in as much as 15% of individuals diagnosed with BC. Currently, due to the absence of the estrogen receptor (ER), progesterone receptor (PR), and human epidermal growth factor 2 (HER2) receptor, there is no hormone-based therapy for TNBC. In addition, there are still no FDA-approved targeted therapies for patients with TNBC. TNBC treatment is challenging owing to poor prognosis, tumor heterogeneity, chemotherapeutic side effects, the chance of metastasis, and multiple drug-resistance. Therefore, various bio-inspired tumor-homing nano systems responding to intra- and extra- cellular stimuli are an urgent need to treat TNBC patients who do not respond to current chemotherapy. In this review, intensive efforts have been made for exploring cell-membrane coated nanoparticles and immune cell-targeted nanoparticles (immunotherapy) to modulate the tumor microenvironment and deliver accurate amounts of therapeutic agents to TNBC without stimulating the immune system.

## 1. Introduction

Breast cancer (BC) is the most prevalent malignancy in females, diagnosed in 24.2% of cases and contributing to a mortality of 15% worldwide [1,2]. Among the different criteria for BC classification, the expression or deficiency of specific receptors: human epidermal growth receptor 2 (HER2), progesterone receptors (PR), and estrogen receptors (ER) are commonly considered [3]. The term triple negative breast cancer (TNBC) stands for a subtype of BC that is negative for the HER2 amplification, and both ER and PR expression [4]. TNBC is a type of BC that is not sensitive to current hormone-based chemotherapies or anti-HER2 treatments [5,6]. In addition, other characteristics of TNBC include a dense extracellular matrix and an increased tumor interstitial fluid pressure (IFP), both of which cause insufficient cancer cell internalization and deep interstitial penetration [7].

The TNBC subtype accounts for 15% of currently diagnosed breast carcinomas and approximately 10–20% of all invasive BC [8]. The diagnosis of TNBC is more common among females aged 40 years or younger. Furthermore, black women are at a higher risk for this type of BC compared with white women [9]. TNBC is usually diagnosed with large tumors, often in a high grade, with the nearby lymph nodes involved. Compared with other BC subtypes, distant recurrence and poor prognosis occur more frequently in patients with TNBC, although women with this disease respond favorably to neoadjuvant chemotherapy [10]. Moreover, approximately 30% of patients who develop metastatic TNBC survive for 5 years [10], with a 13–18 month of median overall survival (OS), given the current treatment regimens for the disease. Understanding the molecular classifications of TNBC can improve treatment selection for different patients [11]. Considering the pathologic, molecular, and clinical aspects, TNBC is one of the most heterogeneous subtypes of BC [12]. Visceral metastasis with a poor prognosis frequently occurs in patients with TNBC [13]. In histopathology, TNBC is considered a heterogeneous disease owing to a high incidence of rare subtypes including adenoid cystic (90–100%), medullary (95%), metaplastic (90%), and apocrine (40–60%) cancers. At the molecular level, heterogeneity can be explained by classifying TNBC into different subtypes, including basal-like (basal-like 1 (BL1) and basal-like 2 (BL2)), mesenchymal (M), mesenchymal stem cell-like (MSL), immunomodulatory (IM), luminal androgen receptor-like (LAR), and unclassified, which are all identified by gene expression analysis [14]. Such heterogeneity not only makes targeting this disease difficult, but it also affects its management. Despite the initial chemosensitivity of TNBC as compared with other types of BC, a high risk of recurrence in patients not achieving pathological complete response (pCR) exists [13]. Treatment re-challenging in recurrent or metastatic TNBC is the only available strategy because of multiple limiting factors such as a poor objective response, multi-drug resistance, and unacceptable toxicity, which all require an urgent need to targeted therapy [6,13].

## 2. Current Strategies for TNBC Treatment

Despite poor prognosis of TNBC with only a small number of known actionable biomarkers, cytotoxic chemotherapy remains the most influential treatment in the neoadjuvant, adjuvant, and metastatic settings [6,15]. While patients with other subtypes of BC display a lower response to chemotherapy, patients with TNBC experience a higher likelihood of responding to chemotherapy, but this anticipatory response to chemotherapy is accompanied by a high risk of recurrence [6,15] notably earlier than other subtypes [16]; this phenomenon is referred to as the TNBC paradox [15]. Standard pre-operative regimens for TNBC that have resulted in the most considerable improvement in pCR include anthracyclines such as doxorubicin (adriamycin, DOX) or epirubicin, alkylating agents such as cyclophosphamide, and members of the taxane family. These drugs are administered concurrently with docetaxel (DTX), or sequentially, such as the addition of DTX or paclitaxel (PTX) to DOX and cyclophosphamide (AC) chemotherapy [17]. Following neoadjuvant chemotherapy, capecitabine is reported to be effective as an adjuvant therapy in patients with residual invasive TNBC [18]. In the case of taxanes, nab-PTX (abraxane), albumin-bound PTX-nanoparticles (NPs) are used to overcome the limitations of traditional PTX. FDA approved the combination of atezolizumab and nab-PTX for the treatment of TNBC [19]. Its new nanosized formulation is attractive owing to its improved pharmacological characteristics, such as linear pharmacokinetics and tolerability, with a higher maximum dose. In addition, compared with PTX, the nab-PTX formulation contains no solvents; therefore, there are no solvent-related hypersensitivities or a need for pre-medication with a corticosteroid or antihistamine. By taking advantage of this solvent-deficient formulation, dosing and toxicity-related issues are all solved [20]. Recently, the FDA approved sacituzumab govitecan (Trodelvy) as an antibody-drug conjugate (ADC) in metastatic TNBC. Sacituzumab govitecan is comprised of SN-38, the active metabolite of irinotecan, conjugated through a linker to the humanized RS7 antibody targeting trophoblast cell surface antigen 2 (Trop-2), a glycoprotein that is overexpressed in TNBC [21].

Additionally, current chemotherapeutic regimens do not produce a satisfactory response because of the genetic heterogeneity and recurrence of TNBC. The 60–70% of patients with TNBC who do not achieve complete response to chemotherapy would benefit from personalized and tumor-specific molecular targeting therapies because of tumor heterogeneity. Approximately 90% of patients with TNBC with persisting sickness after chemotherapy can benefit from drugs targeting different molecular pathways, including the PARP and PI3K inhibitors [6,15]. According to a retrospective analysis, neoadjuvant regimens consisting of taxane and anthracycline resulted in a pCR rate as high as 52% in patients presenting with BL1 tumors, whereas the lowest pCR rates, with 0%, 10%, and 23% in patients with BL2, LAR, and MSL tumor types, respectively [15]. Patients with BRCA1 mutations have shown considerable sensitivity to platinum anti-cancer therapies while showing a lower sensitivity to taxane [22].

The unique characteristics of TNBC, along with a dissatisfying effectiveness of the current treatments [7], are the driving forces to investigate new strategies for the effective treatment of TNBC [22]. Enhanced permeability and retention (EPR) effect is a phenomenon that allows NPs to accumulate in tumor sites [23] and active drug targeting provides specific drug/drug carrier and target cell interaction [22,24]. While, complexities, such as tumor heterogeneity, physiological barriers, and tumor microenvironments (TME) have limited the bench-to-bedside translation of NPs. To provide superior biocompatibility and robust targeting towards desired tissues, biomimetic design of NPs has received increased attention in both research and preclinical studies [25]. Despite all attempts, TNBC is the only BC subgroup that does not have any FDA-approved targeted therapeutic strategy [15,26]. Among many ongoing clinical trials, the study NCT03961698 in phase II trials is investigating the safety and efficacy of the anti-TNBC efficacy of a therapy by inhibiting immune-suppressive tumor-associated myeloid cells by targeting phosphoinositide-3-kinase (PI3K)-γ [27]. In this review, we aim to summarize the currently investigated bio-inspired tumor-homing nanosystems for TNBC.

## 3. Bio-Inspired Tumor-Homing Nanosystems for TNBC Treatment

Different varieties of NPs have been developed to improve their anti-cancer efficacy through the enhanced permeability and retention (EPR) effect and targeted therapy of cancer [6,28,29]; however, most exogenous NPs synthesized using organic or inorganic materials are immunogenic [6,30]. These exogenous NPs are rapidly cleared by the reticuloendothelial system (RES), and exhibit long-term toxicity, poor biodegradation, and low bioavailability [31]. Furthermore, the in vivo off-target effects of chemically modified ligands minimize the tumor selectivity of NPs [6,30]. However, bio-inspired strategies are a novel and attractive subject for drug delivery systems because the designing of NPs is based on mimicking various cellular functions [31]. The TME has unique hallmarks which are distinguishable from normal cells including a low pH, enhanced interstitial fluid pressure (IFP), increased reactive oxygen species (ROS) levels, and the presence of specific enzymes [32]. In addition, cancer immunotherapy improves the immune system’s ability to eliminate cancer by targeting immune cells surrounding the cancer cells. Therefore, engineering NPs that can release chemotherapeutic drugs in the TME or synthetic NPs that are camouflaged within body cells are attractive therapeutic strategies to reduce the off-target effects as well as overcome RES clearance in vivo [31].

### 3.1. Cell Membrane-Coated NPs

Recently, the cell membrane coated nanoparticles due to biomimetic platform, have become an attractive therapeutic strategy for drug delivery applications [33,34]. Coating NPs with membranes of natural cells, such as macrophages, leukocytes, erythrocytes, stem cells, dendritic cells (DC), and tumor cells have been developed to mimic the physiological features of these cells (Figure 1) [33]. Cell membrane-coated NPs contain core–shell structure in which the NP is the core, and the cellular membrane is the shell. The core of the NPs compromised the payload which requires to be delivered to the target site [35]. Cell membrane-coated NPs can be fabricated through top-down fabrication [25] in which the active pharmaceutical ingredients-loaded NPs are prepared by conventional methods, such as emulsion solvent evaporation or self-assembly procedure [25]. To prepare the outer layer, the cells are pretreated with hypotonic cell lysis or mild mechanical stresses including homogenization or sonication then, the proteolipid vesicles of cell membranes are purified and isolated from cells by multiple ultracentrifugation procedures [25,35]. Cell membranes are coated onto NPs through membrane proteins that are present on the membranes of originated cells [35]. Finally, the surface of the cell membrane-coated NPs can be designed by extrusion or electrostatic attraction [25].

In natural-cell mimetic particles as a new delivery system, the core of NPs has a prolonged systemic circulation half-life and increased anti-tumor activity with no reduction in drug loading capacity [36]. These hybrid systems take advantage of both fields, natural cells and synthetic particles, which as natural-coated systems are more functional and complex than the synthetic material alone; further, they are adaptable and tailorable NP delivery systems [37]. Moreover, enhanced cancer-targeted drug delivery is indicated as a satisfactory result of natural mimicking nanoplatforms such as bone marrow-derived mesenchymal stem cell membrane-coated gelatin nanogels (SCMGs), which were developed by Gao et al. [38]. Furthermore, Yang et al. constructed a type of DOX-loaded poly (lactic-co-glycolic acid) (PLGA) NPs coated by human umbilical cord-derived mesenchymal stem cell membrane in order to explore anti-cancer efficacy in HeLa cervical cancer cells and major histocompatibility complex (MHC)C97H liver cancer cells [39,40]. Tian et al. applied PTX-loaded PLGA NPs coated with stem cell membrane vesicle (SCV) to construct SCV-PLGA-PTX and investigated their anti-cancer efficacy against TNBC [33]. In their cell viability study, SCV-PLGA-PTX compared with free PTX and PLGA-PTX showed higher toxicity against 4T1 cells for efficient cancer treatment. In addition, the IC_50_ value of SCV-PLGA-PTX (0.48 μg/mL) was much lower than that of free PTX (1.62 μg/mL) and PLGA-PTX (1.29 μg/mL), indicating an improved tumor targeting efficacy of the coated SCV. Regarding the in vitro study in 4T1 cells, SCV-PLGA-PTX displayed notable improvements in terms of in vitro cancer cell targeting and PTX controlled release. In orthotopic 4T1 tumor-bearing female BALB/c mice, SCV-PLGA-PTX demonstrated the prolonged retention and enhanced accumulation at tumor sites because of the immune escape and cell-mimicking cancer-targeting capacity. The tumor inhibition rates of SCV/PLGA/PTX (78.4 ± 10.6%) were much higher than of free PTX (32.0 ± 6.7%) or PLGA/PTX (53.6 ± 8.3%). SCV-PLGA-PTX dramatically suppressed the tumor growth by enhanced apoptosis and necrosis of tumor cells and reduced the side effects of PTX in a 4T1 tumor model [33]. Therefore, SCV-coated PLGA NPs compared to conventional PLGA NPs due to the better targeting, long retention and more accumulation have demonstrated the superior efficacy against TNBC.

Macrophage and red blood cell (RBC) membranes, in addition to stem cell membranes, are also increasingly used as nature-inspired moieties [36]. Among all camouflaging agents, RBCs seem to be the best choice because of their longer circulation, biocompatibility, and the presence of naturally-occurring functional proteins. Liang et al. constructed a black phosphorous quantum dot (BPQD) that was coated with RBC membrane (RM), forming a biomimetic BPQD-RM nanovesicle (BPQD-RMNV) [41]. BPQD-RMNV was employed with programmed cell death-1 antibody (PD-1 Ab) as a photothermal cancer immunotherapy against TNBC. Different formulations were injected to the 4T1 cell-bearing tumors, irradiated with near infra-red (NIR) laser after 2 h, and subsequent administration of PD-1 Ab to the mice by injection every three days. Tumor growth was estimated by evaluating the bioluminescence signal and tumor volume. The group treated with BPQD-RMNVs + laser + PD-1 Ab showed the best anti-tumor effects compared to other groups, as indicated by the tumor bioluminescence signals, tumor growth rate, and tumor weight. Intensive CD11c^+^ DC recruitment was seen in the primary tumor treated with BPQD-RMNVs + laser groups, indicating the recruitment of DC to uptake tumor antigens. Photothermal therapy of BPQD-RMNV plus NIR irradiation induced tumor cell apoptosis and necrosis, in which cell debris recruit DC and present tumor antigens to native T cells. Furthermore, PD-1 Ab was employed as a checkpoint inhibitor to play an inhibitory role in CD8^+^ T cell exhaustion [41]. Zhai et al. constructed a cytotoxic T lymphocyte (CTL)-inspired nanovesicle (MPV) with a methylene blue (MB) and cisplatin (Pt)-loaded gelatin nanogel core and a RBC vesicle (RV)-derived shell [42]. Applying MPV along with NIR irradiation was part of a combinatorial design to induce TNBC cell death by photochemotherapy with a chance of tumor photoacoustic imaging [42,43]. According to in vitro studies in 4T1 cells, MPV plus NIR irradiation displayed a significant accumulation in tumor cells, showing no MB accumulation in the absence of NIR treatment. Photothermal therapy of MPV plus NIR by thermal ablation released Pt and MB, affected the physicochemical properties of the membrane shell and delivered them into the cytosol of 4T1 cells, which were monitored by imaging of MB-derived fluorescence. In vitro studies revealed the incubation time and irradiation dependency of cell death in both MPV-treated 4T1 and MDA-MB-231 cells. Furthermore, an injection of MPV in 4T1 tumor-bearing mice, which was followed by irradiation after 8 h, displayed the highest temperature in tumors as well as a significant tumor accumulation compared to other groups. In vivo studies have shown that MPVs plus NIR irradiation were capable of the most tumor regression (51.8 ± 14.5%) and the highest level of pulmonary metastasis inhibition (97.7 ± 2.0%) with no significant toxicity [42]. In another biomimetic study by Sun et al. 4T1 cancer cell membrane vesicles (CMV) were utilized as the coating membrane of DOX-loaded gold nanocages (AuNCs) and formed CD-AuNCs. CD-AuNCs plus NIR irradiation selectively delivered DOX to 4T1 cells as a photochemotherapy. The homotypic targeting and NIR irradiation release of DOX were of the interest in 4T1 cells and 4T1 orthotopic mammary tumor models. Based on the in vitro studies, CD-AuNCs with NIR irradiation displayed the highest cellular internalization in a time-dependent manner compared to CD-AuNCs and other groups. Moreover, the IC_50_ value of cells incubated with CD-AuNCs and then irradiated by NIR was the lowest among other formulations, even 3.7-fold lower than CD-AuNCs-treated cells, indicating the effect of induced hyperthermia by NIR laser and the phototoxicity of AuNCs. In cell viability assays, almost all 4T1 cells were dead when treated with CD-AuNCs and NIR irradiation. The in vivo studies on 4T1 orthotopic mammary tumor models displayed the ability of CD-AuNCs to dramatically inhibit the tumor volume and the number of metastatic nodules by about 98.9% and 98.5%, respectively. Aside from the significant anti-cancer results, no important weight loss or organ toxicity was observed [44]. Thus, in vitro and in vivo studies have demonstrated that NPs coated by natural cellular membrane due to the immune escape and cancer cell-mimicking targeting have higher potency, long retention, and more accumulation in the TME.

### 3.2. Immune Cell Targeted Nanosystems

The host’s immune system can be stimulated against the tumor cells by targeting the immune cells or their signaling pathways in the TME (Figure 2) [45]. Clinical data have demonstrated that immunotherapy can impart durable benefits against cancer by targeting the host immune cells and subsequent induction of the patient’s immune system [46,47,48,49]. Cancer immunotherapy through tumor-infiltrating lymphocytes (TILs) is an individualized treatment. TILs have a higher capability of killing solid tumors such as in BC, renal cell carcinoma, and non-small cell lung cancer. TIL inactivation can be exploited by suppressor cells or signaling pathways [50].

In terms of pathways contributing to TIL inactivation, PD-1 receptor, which is expressed on TILs, and its ligand, programmed death-ligand 1 (PD-L1), located on the surface of tumor cells, are the crucial points of interest in immune checkpoint-inhibiting therapies [46]. Targeting and subsequent blocking of such interactions by monoclonal antibodies is a promising paradigm to reactivate immune surveillance in TNBC, based on reported clinical efficiency [50]. Webb et al. synthesized targeted multi-branched gold nanoantennas (MGN) in order to demonstrate their theranostic ability in MDA-MB-231 cells [51]. Functionalization of MGNs took place through the conjugation of human CD247 (B7-H1, PD-L1) antibody and human epidermal growth factor receptor (EGFR) antibody. Antibodies conjugated on MGNs were applied to detect PD-L1 and EGFR in TNBC cell line, which resulted in receptor binding, laser irradiation, light to heat conversion, and ultimately photothermal induced cell death. In addition to individualized immunotherapy, these theranostic MGNs avoid off-target toxicities [51]. TNBC exhibits PD-L1 upregulation on cell surfaces [52], and currently the safety and efficacy of intravenous single-agent pembrolizumab (10 mg/kg every 2 weeks) as an anti-PD-L1 Ab is undergoing phase Ib trials in patients with advanced PD-L1-positive TNBC, gastric cancer, urothelial cancer, and head and neck cancer [52].

Glycosylation of antibodies regulates protein stability [53], activity, and contributes to immunosuppression [50]. Therefore, targeting antibody glycosylation is a potential therapeutic strategy to improve immune checkpoint inhibitors. EGF that induces PD-L1 and PD-1 interaction, requires b-1,3-*N*-acetylglucosaminyl transferase (B3GNT3) expression in TNBC. Manipulating glycosylation by B3GNT3 downregulation blocks PD-1/PD-L1 interactions that indirectly reactivates the anti-tumor immunity of cytotoxic T cells. Antibody targeting of glycosylated PD-L1 (gPD-L1) suppresses PD-L1/PD-1 interaction and promotes anti-TNBC immunity [50]. Li et al. generated an STM108 antibody (anti-gPD-L1 antibody) conjugated with a potent anti-mitotic drug monomethyl auristatin E (MMAE) as gPD-L1 antibody-drug conjugate (gPD-L1 ADC) against TNBC [50]. gPD-L1 ADC selectively suppressed human PD-L1 antigen expressed on BT-549 and MDA-MB-231 cells. An in vivo assay using syngeneic mouse models bearing 4T1 cells demonstrated that gPD-L1 ADC displayed a significant anti-cancer efficacy through induced cell death and a higher survival compared to gPD-L1 Ab alone [50].

Furthermore, tumor associated macrophages (TAMs) represent the largest population of innate immune cells recruited to the TME [54] and have two distinct phenotypes, M1 and M2 (Figure 3) [22]. As a simplified paradigm, the polarization and differentiation of TAM into the cancer-inhibiting M1 and cancer-promoting M2 types represent the two phenotypes of macrophages in the TME [55]. M1 are “pro-inflammatory” cells that are considered as the host defense mechanism against pathogens. M1 macrophages are stimulated by lipopolysaccharide (LPS) and IFN-γ to induce a phenotype that is interleukin 12 ^(^IL-12)^high^, interleukin 6 (IL-6)^high^, inducible nitric oxide synthase (iNOS)^high^, and tumor necrosis factor alpha (TNF-α)^high^ [56,57]. However, M2 macrophages are considered “anti-inflammatory” as they promote tissue repair through cytokine and prostaglandin signaling and result in an IL-10^high^, IL-12^low^, transforming growth factor beta (TGF-β)^high^ phenotype. M2 macrophages promote tumor growth through suppression of cytokine production, reduce activation of T-cells, decrease antigen presenting ability, promote angiogenesis, induce extracellular matrix remodeling, and enhance metastasis-promoting responses and subsequent cell survival [54,56,57]. It has been hypothesized that initial TAMs are predisposed to have M1 function, but are gradually changed to M2 function as tumors grow [6,54].

This is associated with factors such as IL-4, IL-10, TGF-β, prostaglandin (PGE2), and chemokines that are released by tumor cells in response to the changes in the TME, especially in hypoxic environments [54]. TAMs with M2 status express surface markers such as CD204 (scavenger receptor A), CD163, and CD206 (mannose receptor) [6,58] that could be used as a targeting agent [22]. There are clinical studies demonstrating that a high infiltration of TAMs in TNBC tissues correlates with poor prognosis and a higher chance of metastasis, therefore, TAMs are considered as a valuable target for TNBC therapy [59]. Niu et al. synthesized DOX-AS-M-PLGA-NPs targeting the mannose receptor of TAMs [59]. The DOX-loaded PLGA NPs were constructed and their surfaces were functionalized by mannose as the targeting moiety and acid-sensitive polyethylene glycol (PEG). MMTV-M-Wnt-1 (M-Wnt) TNBC cells were cloned from spontaneous mammary tumors in MMTV-Wnt-1 transgenic mice in a congenic C57BL/6 mice. Due to the crucial role of TAMs in tumor growth and metastasis, and their high density in the orthotopic M-Wnt TNBC tumors of C57BL/6 mice, targeting them with DOX-AS-M-PLGA-NPs revealed a high efficiency to inhibit tumor growth and metastasis. DOX-AS-M-PLGA-NPs due to TAM targeting were significantly more effective than the non-targeting one (DOX-AS-PLGA-NPs) in controlling tumor growth. DOX-AS-M-PLGA-NPs in M-Wnt tumor models displayed a significant decrease in the number of F40/80^+^ (a marker for mouse TAM) cells. A single dose injection and multiple injections resulted in a decline of as much as 50% in the number of M2 macrophages and a maintenance of the reduction in M2 macrophages identified as CD206^+^ cells, respectively. Assessing tumor growth inhibition of M-Wnt tumors according to tumor volume and weight showed a slight increase in tumor volume during treatment as well as the least tumor volume and weight at the end of the study in mice treated with DOX-AS-M-PLGA-NPs rather than DOX-AS-PLGA-NPs or DOX alone. DOX-AS-M-PLGA-NPs not only inhibited tumor growth by decreasing the number of TAMs, but they also acted through the direct effects of DOX released from the NPs on tumor cells [59]. We previously reported the photoimmunotherapeutic effect of cetuximab-targeted gold nanorods (CTX-AuNR) for TNBC treatment. We used TAM and non-TAM embedded TNBC spheroids to investigate the therapeutic efficacy of CTX-AuNR plus NIR irradiation. Although TAM embedded TNBC spheroids demonstrated the tumor drug resistance and subsequent TNBC cell survival in DOX treatment, the cytotoxicity and anti-tumor results demonstrated that the efficacy of CTX-AuNR plus NIR irradiation was not significantly different in TNBC cells with or without TAM. CTX-AuNR with NIR irradiation by inducing ROS generation, antagonized tumor hypoxia in TAM-embedded TNBC spheroids, and reprogrammed TAM to the M1 anti-tumor phenotype, as indicated by downregulation of CD206 as a M2 macrophage marker. Thus, CTX-AuNR plus NIR via modulation of the TAM phenotype and the TME demonstrated the successful therapeutic strategy for EGFR-overexpressing TNBC cells [6].

The aggressive nature of TNBC cells is closely dependent on the TME. Cancer-associated fibroblasts (CAFs) are the predominant stromal cell type in TME and via interacting with other stroma cells promote cancer fibrosis and tumor metastasis behavior [62,63]. CAFs are characterized by overexpression of alpha-smooth muscle actin (α-SMA) and fibroblast activation protein (FAP), which play a critical role in angiogenesis, tumor growth, and chemo-resistance [62,63]. Both CAFs and TAMs accelerate cancer progression. Zhou et al. investigated the precise interaction mechanisms and cross-talk between CAFs and TAMs [62]. They examined the synergistic relation of CAFs and TAMs in TNBC progress. The immunohistochemical staining of α-SMA and FAP were used to distinguish CAFs among other stroma cells in 278 TNBC patients while the polarized TAM phenotype was identified by CD163 overexpression. The clinicopathological characteristics among all patients were investigated. CAFs-associated biomarkers (α-SMA and FAP) were overexpressed in TNBC patients with aggressive situation, including recurrence and poor histological differentiation. High activation of CAFs was positively correlated with enhanced infiltration of polarized CD163-positive TAMs and lymph node metastasis in TNBC patients. They have demonstrated that the activation of CAFs, TAMs infiltration, and lymph node metastasis were independent prognostic factors for disease-free survival in TNBC patients. They have concluded that CAFs were associated with infiltration of CD163-positive TAM and lymphatic metastasis and may be potential prognostic predictors of TNBC [62]. Therefore, the cross-talk between CAFs and TAMs are correlated with poor prognosis of TNBC. Accordingly, CAFs by producing a plethora of chemokines, cytokines, extracellular matrix (ECM) proteins, and growth factors can induce TAMs infiltration and consequently promote TNBC metastasis.

Collagen, the most abundant ECM protein in the TME, is produced by fibroblasts. Collagen upregulation and desmoplasia/fibrosis in tumors are associated with enhanced metastatic behavior. TGF-β ligands, enriched in the TNBC TME and produced by CAFs, induce the accumulation of fibrotic desmoplastic tissue and accelerate the cancer progression [64]. In addition, TNBC is susceptible to develop metastases with poor prognosis in central fibrosis situations [64]. Takai et al. have evaluated that targeting the CAFs with Pirfenidone (PFD), an anti-fibrotic agent and a TGF-β antagonist is capable to reduce TNBC metastasis [64]. In patient-derived TNBC xenograft collagen accumulation, TGF-β signaling, and developed lung metastasis were reported. TNBC xenograft tumors, 4T1 TNBC homograft tumors, and tumor specimens of TNBC patients have demonstrated the enhanced CAFs infiltration. CAFs induced primary tumor growth with more fibrosis and TGF-β activation and lung metastasis in 4T1 mouse model. Moreover, they assessed the effects of PFD in vitro and in vivo. They have shown that PFD had inhibitory effects on cell viability and collagen production of CAFs in 2D cell culture. Moreover, CAFs enhanced tumor growth and PFD inhibited the tumor growth induced by CAFs by increasing apoptosis in the 3D co-culture of CAFs-embedded 4T1 tumor cells. In vivo, PFD alone inhibited TGF-β signaling and tumor fibrosis but did not inhibit tumor growth and lung metastasis. However, administration of PFD with DOX inhibited tumor growth and lung metastasis synergistically [64]. Therefore, PFD has great potential for a novel clinically applicable TNBC therapy that targets tumor-stromal interaction and targeting the desmoplasia/fibrosis and TGF-β signaling in TNBC could be of value.

Success in immunotherapy already apparents in many types of malignancies, especially in treating non-small cell lung cancer, pancreatic cancer, BC and melanoma by checkpoint inhibitors. Non-responders to checkpoint inhibitors probably require alternative immunotherapeutic strategies. Each tumor, even with similar histology, may need a unique immunotherapy strategy. Thus, delineating the pathology and underlying mechanisms of tumor evasion is needed to suggest the most effective cancer immunotherapy.

### 3.3. Smart NPs for TNBC Treatment

Smart drug delivery systems (SDDS) with nanocarriers, namely stimuli-responsive NPs are becoming a suitable replacement for conventional drug delivery systems due to their tumor site-specific distribution, controllable drug release [65], prolonged period of drug retention in the tumor site, and lower off-target drug release and drug toxicities (Table 1) [66,67]. Accordingly, smart nanocarriers release the drug at the tumor site [65] in response to stimuli such as low pH (6.5–6.8) [68], hypoxia [69], ROS [70], and overexpressed enzymes [69], which are more common in cancer cells and TME in comparison to healthy cells (Figure 4) [65,71]. Applying SDDS in a combination with other therapeutic agents in complex and heterogeneous cancers such as TNBC may prove crucial to significantly improve antitumor efficacy [72].

A multi stimuli-responsive peptide-based prodrug with structure-transformable characteristics for simultaneous delivery of Pt, adjudin (ADD), and WKYMVm was constructed by Xu et al. as a combination of chemo- and immunotherapy [73]. This SDDS responded well to matrix metalloproteinase-2 (MMP-2), a weakly acidic pH, and glutathione (GSH) to achieve structural transformation with the advantages of efficient and sustained drug delivery. They developed the 2-(Nap)-FFK_Pt-2TPA-ADD_-PLGVRGGGG prodrug that self-assembled to form spherical NPs (2-NPs) and transformed into rod-like NPs (2-NFs) for co-delivery of Pt and ADD. They further constructed another prodrug named 2-(Nap)-FFK_Pt-2TPA-ADD_-GGGPLGVRG-WKYMVm-mPEG1000 that self-assembled into spherical NPs (3-NPs) and transformed into rod-like NPs (3-NFs) for delivery of Pt, ADD, and WKYMVm. A similar structural transformability, prolonged retention, and higher tumor accumulation have been shown between 3-NPs and 2-NPs. Both 2-NPs and 3-NPs also responded well to overexpressed MMP-2 in TME, transforming into rod-like NPs (2-NFs and 3-NFs, respectively) that were internalized by the cell via endocytosis. It was followed by an instant release of the drug triggered by acidic condition and GSH in cytoplasm and continuous formation of nanofibers with extended drug release and deep tumor penetration. Pt and ADD synergistically induced ROS generation and subsequently enhance hydrogen peroxide formation and increase highly toxic hydroxyl radicals that can promote the immunogenic cell death (ICD) response through apoptotic cell death, endoplasmic reticulum stress, and autophagy mechanisms. The transformed nanofibers themselves stimulated ROS production and induced autophagy. 2-NPs and 3-NPs accumulate in the tumors, undergo a change in shape, and release the drugs, for promotion of the ICD response as characterized by ATP secretion and calreticulin (CRT) exposure. ATP as a chemokine facilitated DC recruitment in tumors and CRT induced the engulfment of tumor-associated antigens by DCs that presented the antigen to T cells through the combination with CD91 and toll-like receptor-4 (TLR-4) on the surface of DCs. Based on in vivo assays in a 4T1 orthotopic model, WKYMVm was released by activating formyl peptide receptor 1 (FPR-1); downstream mechanisms enhance the recruitment of CD8^+^ T cells as well as decrease in Foxp^3+^ Tregs (regulatory T cells), which promote immunity against TNBC and activate signaling between DCs and the dying cancer cells. Moreover, 3-NP displayed the highest level of tumor shrinkage (93.1%) and survival, either in the median value (62 days) or overall (82 days), compared to the 2-NP or control groups. The 3-NPs induced a substantial expression of CD91 and TLR-4 that potentiated tumor antigens presenting to DCs, indicating their effects on the innate immunity response. Enhanced expression of activated caspase-3 (cas-3), as well as the elevated release of TNF-α and interferon gamma (INF-γ), were documented in response to 3-NPs. In addition, it was observed that mice treated with 3-NPs had a lower amount of vascular endothelial growth factor (VEGF)-positive cells aside from the reduced levels of CD31 and MMP-9, demonstrating the potency of 3-NPs on the adaptive and innate immune system against TNBC and its metastasis [73]. A similar structural transformability, prolonged retention, and higher tumor accumulation have been shown between 3-NPs and 2-NPs. Both 2-NPs and 3-NPs also responded well to overexpressed MMP-2 in TME, transforming into rod-like NPs (2-NFs and 3-NFs, respectively) that were internalized by the cell via endocytosis. It was followed by an instant release of the drug triggered by acidic condition and GSH in cytoplasm and continuous formation of nanofibers with extended drug release and deep tumor penetration. Pt and ADD synergistically induced ROS generation and subsequently enhance hydrogen peroxide formation and increase highly toxic hydroxyl radicals that can promote the immunogenic cell death (ICD) response through apoptotic cell death, endoplasmic reticulum stress, and autophagy mechanisms. The transformed nanofibers themselves stimulated ROS production and induced autophagy. 2-NPs and 3-NPs accumulate in the tumors, undergo a change in shape, and release the drugs, for promotion of the ICD response as characterized by ATP secretion and calreticulin (CRT) exposure. ATP as a chemokine facilitated DC recruitment in tumors and CRT induced the engulfment of tumor-associated antigens by DCs that presented the antigen to T cells through the combination with CD91 and toll-like receptor-4 (TLR-4) on the surface of DCs. Based on in vivo assays in a 4T1 orthotopic model, WKYMVm was released by activating formyl peptide receptor 1 (FPR-1); downstream mechanisms enhance the recruitment of CD8^+^ T cells as well as decrease in Foxp^3+^ Tregs (regulatory T cells), which promote immunity against TNBC and activate signaling between DCs and the dying cancer cells. Moreover, 3-NP displayed the highest level of tumor shrinkage (93.1%) and survival, either in the median value (62 days) or overall (82 days), compared to the 2-NP or control groups. The 3-NPs induced a substantial expression of CD91 and TLR-4 that potentiated tumor antigens presenting to DCs, indicating their effects on the innate immunity response. Enhanced expression of activated caspase-3 (cas-3), as well as the elevated release of TNF-α and interferon gamma (INF-γ), were documented in response to 3-NPs. In addition, it was observed that mice treated with 3-NPs had a lower amount of vascular endothelial growth factor (VEGF)-positive cells aside from the reduced levels of CD31 and MMP-9, demonstrating the potency of 3-NPs on the adaptive and innate immune system against TNBC and its metastasis [73].

#### 3.3.1. ROS-Responsive NPs

ROS are oxygen-based compounds containing single or unpaired electrons, including superoxide (O_2_^•^), hydroxyl radical (^•^OH), and nitric oxide (NO^•^), and are highly reactive reagents. Under normal physiological conditions, intracellular ROS are generated as byproducts of mitochondrial processes, metabolism, and enzymatic activity [74]. They are efficiently neutralized by superoxide dismutase, catalase, GSH, and thioredoxin enzymes to control homeostasis of redox-oxidative states. Interestingly, enhanced levels of ROS or deregulated redox-oxidative homeostasis, as a hallmark of cancer cell progression, metastasis, and survival, has been extensively documented.

ROS has a critical role in activating pro-tumorigenic signaling pathways, especially the nuclear factor kappa-light-chain-enhancer of activated B cells (NF-κB) cascade in MMP-mediated tumor cell invasion and metastasis [74]. Considering the role of enhanced ROS levels and associated oxidative stress in DNA and protein denaturation [79,80], which is followed by the proliferation of several types of cancer cells, employing ROS-responsive materials and linkers might be crucial to develop tumor targeted SDDS [79]. As ROS and oxidative stress play critical roles in cancer progression and metastasis, antioxidant therapy is a good therapeutic strategy to neutralize ROS-mediated cellular growth. Shashni and Nagasaki developed ROS scavenging nitroxide radical-containing NPs (RNPs). Among micelle structured RNPs, RNP^N^ is pH-sensitive while RNP^0^ is pH insensitive. The nitroxide radical (2,2,6,6-tetramethylpiperidine-1-yl)-oxyl (TEMPO) is the functional part of the NPs that catalytically removes ROS. The ROS scavenging part of the RNP^N^ is 4-amino TEMPO, which makes these micelles sensitive to acidic conditions. In addition, pH sensitivity is the parameter contributing to the higher scavenging ability of RNP^N^ compared to that of RNP^0^. RNP^N^ and RNP^0^ showed significant inhibitory effects on the proliferation and colony forming potential of MDA-MB-231 and MCF-7 cells, in a dose-dependent manner. RNP^N^ was more effective in inhibiting the metastasis and invasion compared to RNP^0^ in MDA-MB-231 cells. Furthermore, in vivo results demonstrated that IV administration of RNPs in an MDA-MB-231 xenograft model influentially reduced tumor volume and weight and downregulated MMP-2 and NF-κB proteins, while maintaining an almost constant mouse body weight. The cytotoxic effects of RNPs were comparable to those of PTX although they were more significant than the cytotoxic effects observed in TEMPO- and saline-treated groups. Regarding the ROS generation in treated groups, RNP-treated group had the lowest level of ROS in tumors compared to those in PTX-, TEMPO-, or saline-treated groups, with lower levels found in the RNP^0^-treated group than in the RNP^N^-treated group [74]. The aberrant ROS generation and ineffective neutralization of excessive ROS level can cause cancer growth and progression through different signaling pathways including phosphatidylinositol 3-kinase/protein kinase β/the mechanistic target of rapamycin (PI3/Akt/mTOR), VEGF/VEGFR, phosphatase and tensin homolog (PTEN), and MMPs [81]. Therefore, developing a smart synergistic therapy system that can ensure the on-demand release of chemotherapeutics and reduce ROS generation in the presence of low pH could be a successful therapeutic strategy for TNBC.

#### 3.3.2. pH-Responsive NPs

An acidic pH of the solid TME, originating from the Warburg effect, might be considered as the key to modify the structure of NPs to facilitate drug release. Three different mechanisms comprise the acid-responsive drug release. One mechanism is based on the difference between the pK_a_ of the NPs’ constructing moieties and the tumor interstitial pH, which cause a protonation of the functional groups and subsequent particle transformation. Another mechanism is the cleavage of pH-sensitive linkages. The last strategy is to use pH-sensitive insertion peptides, in which an improved cellular membrane penetration in acidic environments was achieved. Fan et al. designed cationic liposome nanocomplex (Combo NCs) prodrug for the co-delivery of DTX and gemcitabine (GEM) characterized as an enzymatic and pH dual-stimulus-responsive release [75]. They developed Combo NCs using a hyaluronic acid-gemcitabine (HA-GEM)-conjugated nanocomplex (NC) comprising 1,2-dioleoyl-3-trimethylammonium-propane (chloride salt) (DOTAP) self-assembled cationic liposomes loaded with DTX. DOTAP is a positively charged phospholipid with an ammonium group that creates cationic liposomes by self-assembly. HA-GEM is anionic under physiological conditions; therefore, HA-GEM conjugates can be attached to DTX-loaded liposomes through electrostatic interactions. Hyaluronic acid (HA) is the targeting agent for CD44 receptors that mediate the Combo NCs’ endocytosis, while HA-GEM shield surrounding Combo NCs, prevent premature DTX leakage. An additional HA coating on the surface of NCs is degraded by hyaluronidases such as Hyal-1 and exposes the inner cationic liposome. An acidic pH in the lysosomal environment caused the cleavage of ester bonds in the HA-GEM conjugate, which facilitated GEM release. To investigate the anti-cancer efficiency of Combo NCs, TNBC cell lines expressing CD44 receptors, including MDA-MB-231 and MCF-7 were used; MDA-MB-231 cells express a significantly higher level of CD44 (10.8-fold) relative to MCF-7 cells. Accordingly, all in vitro assays such as cell cytotoxicity, apoptosis, wound healing studies, as well as cell cycle analyses were employed on MDA-MB-231 cells. Combo NCs showed a significantly high cytotoxicity, apoptotic effect, anti-migration efficacy (wound healing study), and considerable S phase blocking effect in MDA-MB-231 cells. Further studies demonstrated that the combination of DTX and GEM in Combo NCs improve GEM anti-cancer efficacy by influencing the enzymes involved in GEM activation through upregulation of deoxycytidine kinase (dCK), cytidine deaminase (CDA) downregulation, followed by metabolism activation. They have demonstrated that co-delivery of DTX and GEM in Combo NCs by modulating the dCK/CDA ratio to the highest level compared to other groups (15.1-fold compared with blank control) induced the accumulation of GEM, which is responsible for the higher toxicity of Combo NCs compared to other groups. In vivo studies carried out on MDA-MB-231 cell-bearing female BALB/c nude mice indicated that Combo NCs, in comparison with the free GEM and free Combo, show higher anti-tumor efficacy (approximately 93 mm^3^ decrease in tumor volume) and less systemic toxicity owing to their active targeting, suitable DTX/GEM ratio, carriers’ size, and biocompatibility of carriers’ components [75]. Consequently, this formulation through binding to CD44 receptor induced the endocytosis of target cells and after internalization into lysosomes, the acidic condition triggered cleavage of the ester bond [82], causing GEM release; hyaluronidase enzyme caused degradation of HA, exposing the inner cationic core and thus inducing lysosomal escape, resulting in release of lipophilic DTX and hydrophilic GEM which contribute to its synergistic therapeutic efficacy.

#### 3.3.3. Enzyme-Responsive NPs

Transportation across the tumor vessels and through the tumor interstitial matrix is determined by the tumor architecture and NPs characteristics, including size, charge, and configuration [83]. Among these parameters, size plays a key role in boosting both the permeability and retention of NPs in the TME. The penetrating efficiency of NPs are negatively correlated to their size, which means that small NPs have shown enhanced trans-vascular and interstitial transport [83]. However, smaller-sized NPs have a shorter blood half-life circulation, which enhance undesired distribution and other side effects [76,83]. Conversely, relatively larger particles are characterized with a longer blood half-life but do not penetrate as deeply [83]. Thus, for ideal penetration, the initial size of NPs should be large enough to provide long blood circulation and selective extravasation, but once they enter into the TME from leaky vasculature, a smaller size is preferable. The contradictory requirements for size-dominating permeation and retention in the TME have promoted the development of enzyme-triggered size-shrinkable NPs, which have a potential to change their size in the existence of enzymes as a stimulus [83].

Hu et al. designed size-reducible NPs in order to overcome the diffusion obstacles and reduce their rapid clearance in the in vivo TME [76]. The angio-DOX-DGL-GNP consists of gelatin NPs (GNPs) as a core with DOX and angiopep-2 linked with dendrigraft poly-lysine (DGL) in the outer portion. Angiopep-2 attached on the surface of NPs bind with low density lipoprotein-receptor (LRP) overexpressed on TNBC cells, facilitating their uptake as well as their accumulation. A significantly higher cellular uptake of angio-DOX-DGL-GNP than of DOX-DGL-GNP was observed in 4T1 cells, indicating that angiopep-2 targeting facilitated cellular internalization. In addition, angio-DOX-DGL-GNP in the presence of MMP-2 rather than without MMP-2 showed deeper penetration due to their smaller size. The shrinking process of large-sized angio-DOX-DGL-GNP occurred in the presence of MMP-2, which is overexpressed in TNBC cells, causing a size reduction from 185.7 nm to 55.6 nm. Subsequently, the cleavage of the acid-sensitive cis-aconitic anhydride bond between DOX and DGL caused DOX release. In vivo studies demonstrated that angio-DOX-DGL-GNP was efficiently accumulated and penetrated into the tumor tissues of 4T1 tumor-bearing mice. A considerable inhibitory effect (74.1%) of angio-DOX-DGL-GNP was observed in a 4T1 cell-bearing mouse model, which was significantly higher than other treated groups. There was no significant weight gain in the mice and these particles were found to be biologically safe [76]. Ruan et al. have developed a SDDS that provides satisfactory permeation and retention aside from targeting [7]. RRGD, a tandem peptide of RGD (arginyl-glycyl-aspartic acid) and octaarginine, as a targeting agent was conjugated on NPs and the direct delivery system to the extracellular matrix and tumor site. Gold nanoparticles (AuNPs) were conjugated to MMP-2 degradable gelatin NPs (GNPs) that formed G-AuNPs and DOX was attached to AuNPs through a pH sensitive hydrazone bond, creating G-AuNPs-DOX-RRGD (Figure 5). G-AuNPs-DOX-RRGD specifically target 4T1 tumors through RRGD adhesion and binding. After retention in the tumor extracellular matrix, the large G-AuNPs-DOX-RRGD came in contact with MMP-2 protease and formed smaller-sized AuNPs-DOX-RRGD, reducing the particle size from 185.9 nm to 71.2 nm. Moreover, AuNPs released following the G-AuNPs-DOX-RRGD shrinkage by enzymatic exposure improved their penetration. After releasing AuNPs and penetrating deep into the tumors, hydrazine, an acid sensitive linker, was cleaved in the low pH of the TME, releasing DOX conjugated to AuNPs. To investigate the different aspects of tumor penetration and anti-tumor efficacy of the G-AuNPs-DOX-RRGD in comparison with other formulations such as AuNPs-DOX-RRGD, GNPs-DOX-RRGD, G-AuNPs-DOX-PEG, experiments were conducted on 4T1 tumor cells and spheroids as well as in 4T1 tumor-bearing BALB/c mice. Cellular uptake experiments revealed that 4T1 cells were capable of taking up RRGD-decorated particles at much higher rates than that of non-decorated ones (G-AuNPs-DOX-PEG). The cell penetration assay was performed in three-dimensional 4T1 tumor spheroids, where AuNPs-DOX-RRGD showed a stronger fluorescence intensity compared to G-AuNPs-DOX-RRGD, indicating that a smaller particle size penetrated into the deep regions of tumor spheroids. Although G-AuNPs-DOX-RRGD could not diffuse as deeply, pre-treatment with MMP-2 caused a shrinkage of the NPs, allowing a higher density of penetration. Treatments with GNPs-DOX-RRGD showed the same fluorescence intensity in the presence or absence of MMP-2. Furthermore, the distribution of G-AuNPs-DOX-RRGD was much higher than that of G-AuNPs-DOX-PEG, indicating the role of RRGD in penetrating tumor spheroid sections [7]. The penetrating efficiency of NPs are negatively correlated to their size, which means that small NPs have shown enhanced trans-vascular and interstitial transport [83].

However, smaller-sized NPs have a shorter blood half-life circulation, which enhance undesired distribution and other side effects [76,83]. Conversely, relatively larger particles are characterized with a longer blood half-life but do not penetrate as deeply [83]. Thus, for ideal penetration, the initial size of NPs should be large enough to provide long blood circulation and selective extravasation, but once they enter into the TME from leaky vasculature, a smaller size is preferable. The contradictory requirements for size-dominating permeation and retention in the TME have promoted the development of enzyme-triggered size-shrinkable NPs, which have a potential to change their size in the existence of enzymes as a stimulus.

#### 3.3.4. Nitric Oxide (NO)-Responsive NPs

Nitric oxide (NO) is a free radical generated by endothelial cells. It is involved in vasodilation through soluble guanylate cyclase (sGC) enzyme activation, which is then followed by cyclic guanosine-30, 50-monophosphate (cGMP) production [84].

Subsequently, the cGMP-dependent protein kinase causes vasodilation. The feasibility of locally administrated NO donors to tumors was reported with satisfying results. A short half-life, having low payloads, and insufficient organ or tissue selectivity make this radical of an interest in developing suitable delivery systems [77]. Alimoradi et al. used a copolymer of styrene-maleic acid (SMA) to synthesize SMA-DOX micelles through hydrophobic interactions [77]. They then constructed SMA-tDodSNO as a type of locally administrated NO donor (SMA-*tert*-dodecane S-nitrosothiol [tDodSNO]) in order to improve the anti-cancer efficacy of DOX or SMA-DOX (Figure 6). They investigated the enhanced EPR effect of tDodSNO, which was encapsulated into SMA nanomicelles (SMA-tDodSNO). Treatment of 4T1 cells with SMA-tDodSNO (40 μM) compared to SMA-DOX micelles significantly enhanced their cellular uptake and DOX accumulation inside the cells (2-fold; *p* < 0.001). The in vitro anti-cancer efficacy of SMA-tDodSNO nanomicelles were compared with SMA-DOX micelles or free DOX on 4T1 cells. Based on in vitro results, a combination treatment with SMA-tDodSNO significantly reduced the cell viability and increased early and late apoptosis in 4T1 cells by arresting cancer cells in the sub-G1 phase. The molecular mechanism contributing to the cellular damage of SMA-tDodSNO is the reaction occurring between ROS and reactive nitrogen species (RNS) produced by their respective DOX and NO donor. Furthermore, NO production through inhibition of nuclear factor kappa B and Snail is another mechanism that enhanced the sensitivity of cancer cells to the chemotherapeutic effects of DOX. Subcutaneous administration of SMA-DOX to mice xenografted with 4T1 cells have shown that SMA-DOX-treated mice had a similar tumor progression rate as compared to controls the day of formulation injection (5.6-fold), representing the week anti-tumor potency of SMA-DOX (5 mg/kg). However, SMA-tDodSNO caused a 2-fold tumor size reduction compared to the control group (244% vs. 561% for control and 569% for SMA-DOX) without inducing significant toxicity as evaluated by body weight loss. Authors concluded that SMA-tDodSNO synergistically improved the anti-cancer properties of SMA-DOX through the EPR effect, endosomal membrane disruption, and enhanced intracellular DOX accumulation in tumor tissues [77].

Combined treatment of SMA-DOX and SMA-tDodSNO at day 10 post-injection significantly reduced the tumor volume (195%) than SMA-DOX treatment (569%) in mice.

#### 3.3.5. Hypoxia-Activated NPs

Hypoxia plays a crucial role in drug delivery systems. Hypoxia-activated NPs benefit from an oxygen pressure lowering gradient in the tumor region from the surface to the core, in which low oxygen pressure in cancer cells acts as a trigger for drug release in cancer cells but not in normal cells (normoxic cells) [66]. Targeted TNBC therapy, hypoxia-activated chemotherapy, and photodynamic therapy (PDT) are all simultaneously applied by the NPs, which were constructed by Wang et al. [78]. Tumor homing and penetrating cyclic peptide (iRGD)-modified hybrid PLGA/lipid NPs were used to encapsulate indocyanine green (ICG) as the photosensitizer and tirapazamine (TPZ) as the hypoxia-activated chemotherapy (iRGD-PLGA-ICG-TPZ) against TNBC. The shell-forming agents were DPPC (1,2-dipalmitoyl-sn-glycero-3-phosphocholine) as a lipid monolayer-forming agent, DSPE-mPEG (1,2-distearoyl-sn-glycero-3-phosphoethanol-amine-*N*-mPEG) as the NP stabilizing agent, and DSPE-PEG-DBCO (DSPE-PEG-dibenzocyclooctyl) as the conjugating agent. iRGD is a 9-amino acid cyclic peptide used as a tumor-homing and penetrating peptide. The construction was then followed by linking the iRGD as an intratumoral permeability-enhancing agent upon interacting with αv integrins and neuropilin-1. iRGD-PLGA-ICG-TPZ showed significantly improved penetration in both in vitro 3D tumor spheroids and orthotopic 4T1 tumors in vivo. The mechanism by which iRGD-PLGA-ICG-TPZ and laser irradiation induced 4T1 cell death is initiated by PDT through ICG, resulting in a hypoxic environment that triggered TPZ for DNA denaturation and induced selective death of hypoxic TNBC cells (Figure 7). In vivo results have shown fast tumor growth in the blank control, while a moderately restricted tumor growth was observed for iRGD-targeted NPs loaded with a single drug (iRGD-PLGA-ICG or iRGD-PLGA-TPZ), a combination of ICG and TPZ, and the PLGA-ICG-TPZ groups. However, iRGD-PLGA-ICG-TPZ with NIR irradiation successfully delivered ICG and TPZ in 4T1 orthotopic tumors in which primary tumor progression and metastasis were inhibited with minimal side effects [78].

#### 3.3.6. Thermo-Responsive NPs

Higher temperature of tumor tissues compared to normal ones or the external heating of tumor sites is considered a factor to achieve controlled drug release [66]. Altering the temperature is a simple external stimulus for in vitro and in vivo experiments [85,86], and it is also the least invasive therapeutic intervention [85,87]; thus, it is being increasingly applied in SDDS [86]. A kind of merged TNBC therapy applied by Ding et al. consists of photo-, chemo- and gene therapy [72] (Figure 8).

The design of such multi-functional SDDS was based on a two-layer structure: the building block of the core-shell structure was a thermo-sensitive co-polymer named poly-((2-(2-methoxyethoxy)-ethyl methacrylate-co-oligo-(ethylene glycol) methacrylate)-co-2-(dimethylamino)-ethyl methacrylate-b-PLGA NP). Co-encapsulation of this co-polymer with two cytotoxic drugs, PTX and DOX (NP-PD), and surface absorbance of small interfering RNAs against survival (NP-PD-S), formed a nanostructure to be finally coated by a self-polymerized dopamine (PDA) film (NP-PD-S-PDA). The PDA film not only displayed a photothermal therapeutic efficacy, but also represented a protective efficiency against an uncontrollable release of drugs. The in vitro cellular uptake demonstrated that NP-PD-S-PDA was taken up by MDA-MB-231 cells, resulting in a remarkable decline in cell viability and indicating that this is an effective combination therapy. A remarkably reduced cell viability was seen when the cells were treated with NP-PD-S-PDA under laser irradiation for 5 min. After 48 h, the IC_50_ of cells treated with NP-PD-S-PDA plus NIR irradiation decreased to 2.7 ng/mL, about 1/55 of that for NP-PD-PDA, 1/20 of that for free PTX and DOX combination, and 1/15 of that for NP-PD. In agreement with in vitro assays, results achieved from orthotopic female BALB/c nude mice bearing MDA-MB-23, demonstrated how NP-PD-S-PDA played a role as a multifunctional anti-cancer nanocomposite. In tumors treated with NP-PD-S-PDA plus NIR irradiation, PDA generated sufficient heat and the thermal ablation caused NPs collapse, triggering PTX and DOX release within the tumor.

Comparing with mono- (NP-PDA under laser, NP-PD, NP-S) and dual-therapies (NP-PD-S, NP-PD-PDA under laser, NP-S-PDA under laser), NP-PD-S-PDA as a triple regimen demonstrated the most effective anti-tumor effect [72].

Patients with TNBC initially respond to conventional chemotherapy, but cancer recurrence leads to a worse outcome compared to that of other BC subtypes. BC is genetically and phenotypically heterogeneous, therefore, targeting TNBC cells, the TNBC microenvironment, or their signaling pathways regardless of their variation among patients or within the same patient is rather difficult [88]. Thus far, FDA has not approved any targeted therapies for patients with TNBC. Researchers have made great progress in developing novel ADC or NPs with the aim of treating TNBC, and a variety of new therapeutics have been developed to extend the survival of patients with TNBC. Combinations of current conventional therapies, cancer cell therapies, and immune cell-targeted NPs have a potential to overcome TNBC resistance. In addition, cell membrane-coated NPs are viewed as promising cargoes to TNBC treatments and may help in designing strategies to treat resistant tumors. Cell membrane-coated NPs can deliver accurate amounts of therapeutic agents to TNBC without stimulating the immune system. SDDS can trigger drug release in the TME and can deliver drug selectively; therefore, they may be used as potential cargoes for TNBC treatment. We believe that bio-inspired NPs and SDDS will play important roles in the detection and development of personalized medicine for TNBC, and substantially overcome the limitation of current cancer therapies. More studies are required in the areas of smart NPs, gene therapy, and on the development of immune cell-targeted delivery systems for the treatment of TNBC. We believe that such efforts in a relatively short time will result in the discovery of more effective therapies for TNBC [89]. The multi-functional nano drug delivery with molecular-targeting capacity in impaired cancer related pathways are likely to yield good results in clinical trials as they have proved effective in in vitro and in vivo studies.

## Figures and Tables

**Figure 1 pharmaceutics-13-00287-f001:**
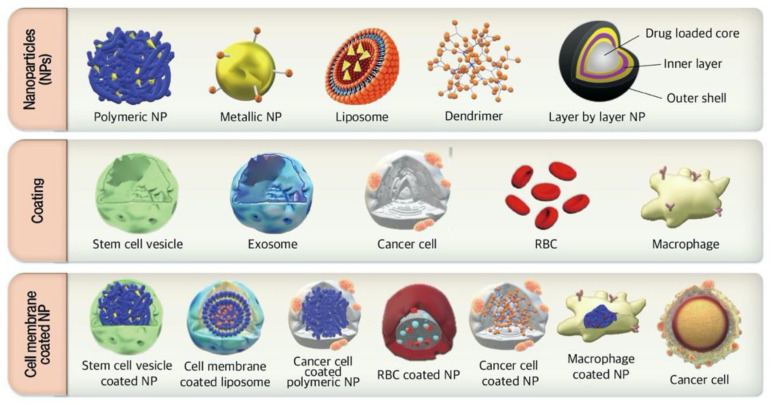
General scheme of membrane-coated nanoparticles (NPs). Different source of cancer cells or immune cells are applied as the shell of NPs by camouflaging different membranes. The membranes of cancer cells or immune cells are isolated from their source cells and extruded to obtain membrane vesicles. The vesicles then fuse with different types of NPs as a core to form membrane-camouflaged NPs. RBC, red blood cell.

**Figure 2 pharmaceutics-13-00287-f002:**
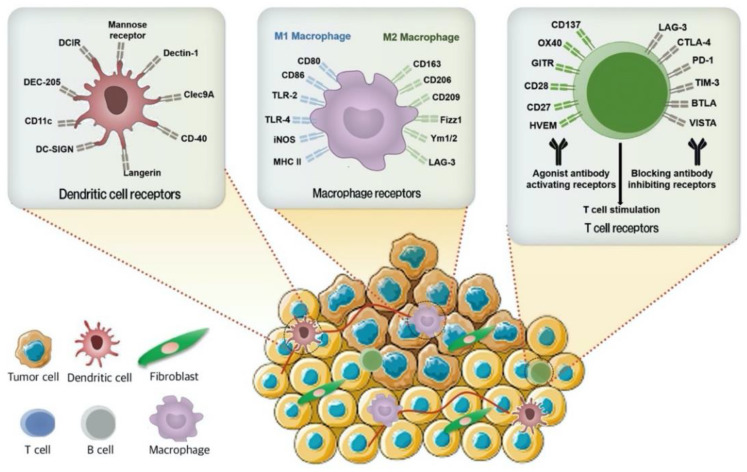
Schematic representation of the surface markers of immune cells including macrophage, dendritic cells, and T cells to design immune cell targeted NPs. Macrophage, dendritic cells, and T cells expressing different cell surface receptors are good therapeutic targets for TNBC therapies by modulation the tumor microenvironment (TME). DCIR, dendritic cell (DC) immunoreceptor; DC-SIGN, dendritic cell-specific intercellular adhesion molecule-3 grabbing non-integrin; DEC-205, CD-205; Clec9A, C-type lectin-like receptor on DC; TLR-2, toll-like receptor 2; TLR-4, toll-like receptor 4; iNOS, inducible nitric oxide synthase; MHC, major histocompatibility complex; Ym1/2, chitinase-like proteins; LAG-3, lymphocyte-activation gene 3; OX40 or TNFRSF4, tumor necrosis factor receptor superfamily member 4; GLTR, GATEWAY-compatible lentiviral tetracycline-regulated RNAi; HVEM, herpes virus entry mediator; CTLA-4, cytotoxic T lymphocyte-associated antigen-4; PD-1, programmed cell death-1 antibody; Tim-3, T-cell immunoglobulin and mucin domain 3; BTLA, B- and T-lymphocyte attenuator; VISTA, V-domain Ig suppressor of T cell activation.

**Figure 3 pharmaceutics-13-00287-f003:**
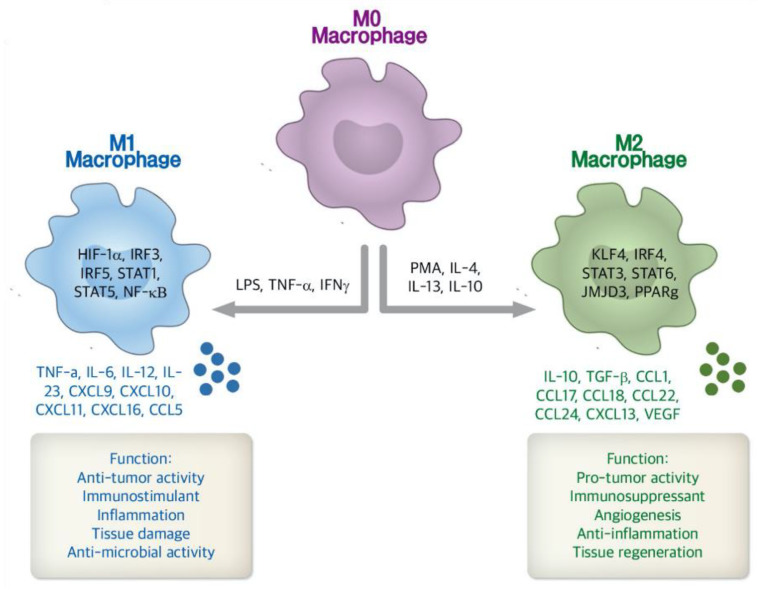
The schematic figure represents the different stimulating factors, cytokines, and biological functions between M1 and M2 phenotypes of macrophages. As a simplified paradigm, TAM differentiation into the cancer-inhibiting M1 and cancer-promoting M2 types represent the two phenotypes of macrophages in the TME. M1, pro-inflammatory cells, are stimulated by lipopolysaccharide (LPS), tumor necrosis factor alpha (TNF-α) and interferon-gamma (IFN-γ) to induce a phenotype that is interleukin 12 (IL-12)^high^, interleukin 6 (IL-6)^high^, inducible nitric oxide synthase (iNOS)^high^, and tumor necrosis factor alpha (TNF-α)^high^. However, M2 macrophages are activated by phorbol myristate acetate (PMA) and IL-4,10, 13. M2 with anti-inflammation characteristics promote tissue regeneration and immunosuppression through cytokine and prostaglandin signaling and result in an IL-10^high^, IL-12^low^, transforming growth factor beta (TGF-β)^high^ phenotype. CCL, chemokine (C-C motif) ligand; CXCL, chemokine (C-X-C) ligand; HIF, hypoxia inducible factor; IRF, interferon regulatory factor; JMJD, Jumonji doman-containing protein; NF-κB, nuclear factor κB; KLF, Kruppel-like factor; PPAR, peroxisome proliferator-activated receptors; STAT, signal transducer and activator of transcription; VEGF, vascular endothelial growth factor [60,61]. Figure is adapted with permission from [61], Springer Nature, 2019.

**Figure 4 pharmaceutics-13-00287-f004:**
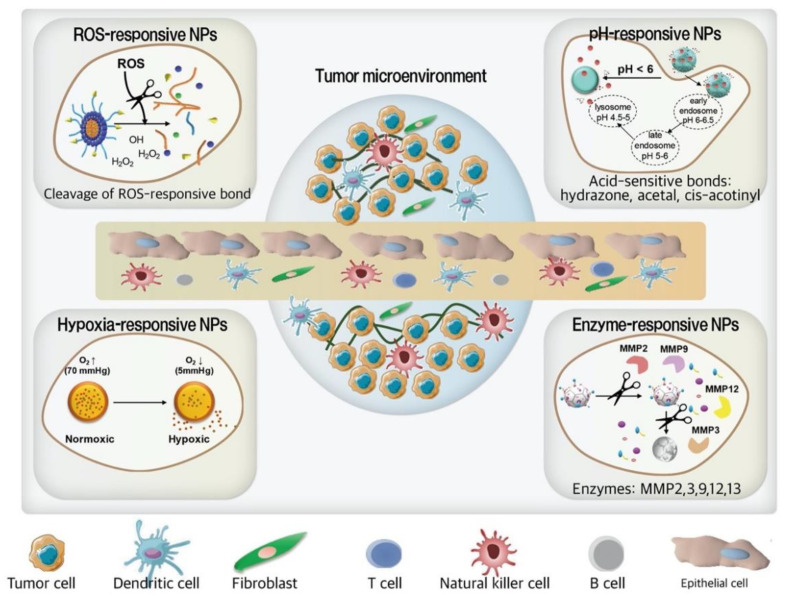
Schematic illustration of tumor microenvironment-responsive smart NPs including pH, ROS, hypoxia, and enzyme-responsive NPs for treatment of TNBC. ROS-responsive and hypoxia activated NPs trigger the drug release in enhanced ROS formation or reduced oxygen in cancer cells. In pH-responsive NPs, acid-sensitive bonds in low pH of TNBC environments are cleaved. Some enzyme-sensitive bonds are easily cleaved by overexpressed enzymes such as MMP in the tumor microenvironment.

**Figure 5 pharmaceutics-13-00287-f005:**
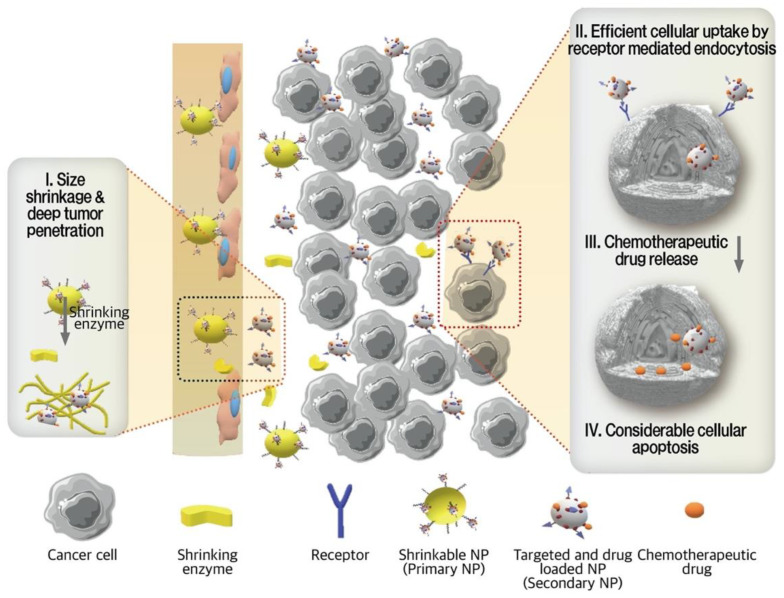
Schematic illustration of enzyme responsive shrinkable NPs. Size transition process of the primary NPs and the following step, drug release, occurred in two stages. Surround the perivascular sites, large-sized particles experienced an MMP2 enzyme enrolled size shrinkage and changed into smaller particles to penetrate deeply in tumors. High cellular uptake of the reduced-sized particles took place through receptor-ligand interactions. Within the acidic pH of cancer cells, the acid-labile bonds were cleaved, and the drug carrier released its cargo finally, cell apoptosis was induced by chemotherapeutic drug.

**Figure 6 pharmaceutics-13-00287-f006:**
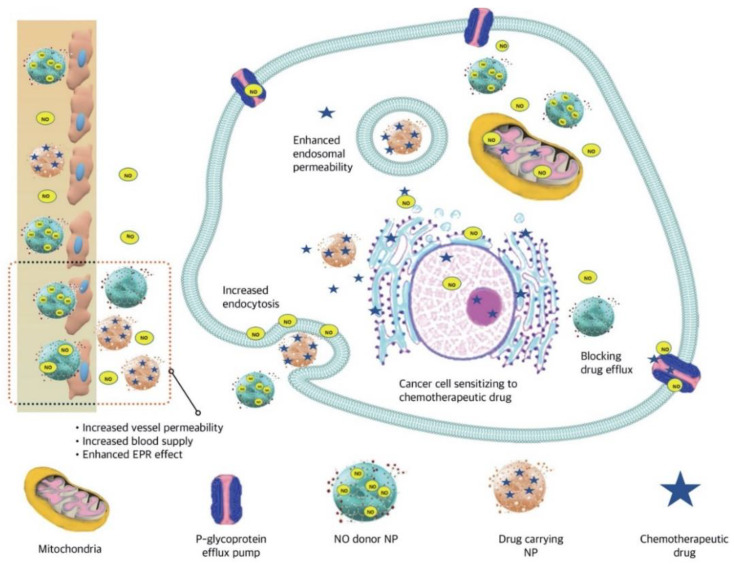
Schematic illustration of combined delivery of intracellular RNS producing and chemotherapeutic delivering NPs. Concurrent use of NO donor particles with NPs delivering chemotherapeutic drug enhanced blood supply and tumor penetration of the NPs. NO released from its donor accelerated endocytosis, enhanced membrane permeability and endosomal escape as well. Intracellular accumulation of both particles followed by the production of ROS and RNS, resulted in depolarization of mitochondrial membrane. Moreover, decrease in drug efflux and resistance through Pgp was another anticancer property of this combination delivery system.

**Figure 7 pharmaceutics-13-00287-f007:**
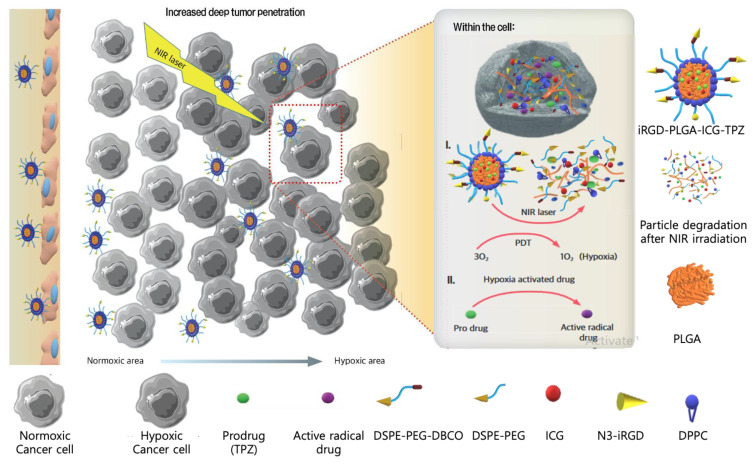
Schematic illustration of hypoxia activated drug delivery system (iRGD-PLGA-ICG-TPZ) via tumor targeting moiety, accompanied with photodynamic therapy (PDT). Targeted nanoparticles (NPs) penetrated to the deepest parts of the tumor, hypoxic areas. PDT of photosensitizer plus near infra-red (NIR) irradiation provided an environment less in oxygen by consuming O_2_ and producing reactive oxygen species (ROS). In addition, PDT caused NP structure degradation to release hypoxia-responsive prodrug. Intensified hypoxic areas triggered the activation of prodrug into radical chemotherapeutic drugs. iRGD, penetrating cyclic peptide; TPZ, tirapazamine; PLGA, poly (lactic-co-glycolic acid); DPPC, 1,2-dipalmitoyl-sn-glycero-3-phosphocholine; PEG, polyethylene glycol; DPPC, 1,2-dipalmitoyl-sn-glycero-3-phosphocholine, DSPE, 1,2-distearoyl-sn-glycero-3-phosphoethanol-amine; DBCO, dibenzocyclooctyl; ICG, indocyanine green.

**Figure 8 pharmaceutics-13-00287-f008:**
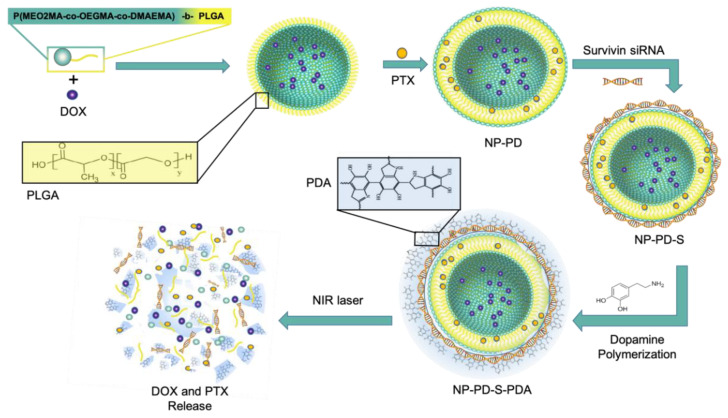
Schematic representation of constructing P(MEO2MA-co-OEGMA-co-DMAEMA)-b-PLGA based NP which delivers doxorubicin (DOX) and paclitaxel (PTX) simultaneously (NP-PD). The surface absorbance of small interfering RNAs against survival (NP-PD-S), formed a nanostructure to be finally coated by a self-polymerized dopamine (PDA) film (NP-PD-S-PDA). Drug release took place following near infra-red (NIR) laser-triggered NP collapse.

**Table 1 pharmaceutics-13-00287-t001:** Overview of the TME-responsive smart NPs for TNBC treatment.

Nanoparticles	Stimuli	Carrier Type		Bioactive Compound	Ligand	Target	TNBC Cell Line	Outcome	Ref.
Multi stimuli NP	MMP-2 GSH Acidic pH	2-(Nap)-FFK_Pt-2TPA-ADD_-PLGVRGGGG(2-NPs) 2-(Nap)-FFK_Pt-2TPA-ADD_-GGGPLGVRG-WKYMVm-mPEG1000(3-NPs)		Pt ADDWKYMVm	WKYMVm	FPR-1	MDA-MB-231 4T1 MCF7ADR orthotopic model	Highest cell death in all three cell lines by both NPs No body weight loss in both NPs treated mice Well tolerable tumor inhibitory effects Prolonged tumor retention 93.1% tumor shrinkage in 4T1 model 2.7- fold increase of overall survival	[73]
ROS-responsive NP	ROS scavenging Acidic pH	RNP^N^ pH-sensitive	MeO-PEG-*b*-PMNT	TEMPO	-	ROS	MDA-MB-231 xenograft model	61% MDA-MB-231 cell viability treated with RNP^0^ 98% MDA-MB-231 cell viability treated with RNP^N^ Considerable anti-migratory effect on MDA-MB-231 cell Higher invading inhibition potential for RNP^N^ rather than RNP^0^ Significant anti-tumor effect and tumor weight decrease Important downregulation of MMP-2 and NF-κB in tumor Insignificant adverse effects on mouse body weight	[74]
		RNP^0^pH-insensitive	MeO-PEG-*b*-PMOT						
pH-responsive NP	Acidic pH	DOTAP		DTX GEM	HA	CD44	MDA-MB-231	Highest induced apoptosis: 80 ± 5.12% Strongest anti-migration effects in MDA-MB-231 cell line by Combo NCs Almost 93 mm^3^ decrease in tumor volume in MDA-MB-231 tumor bearing mice Lack of considerable systemic toxicity in Combo NCs treatment	[75]
Enzyme-responsive shrinkable NP	MMP2 enzyme NIR	G-AuNPs		DOX	RRGD	Extracellular matrix	4T1 cells bearing mice	Improved tumor targeting Deep tumor penetration (75.5%) Enhanced tumor accumulation Acidic environment dependent drug release No considerable pulmonary metastasis Displaying the high tumor growth inhibition	[7]
		GNP with drug loaded DGL		DOX	Angiopep-2	LRP_1_	4T1 cells bearing mice	Higher cellular uptake due to efficient targeting Considerable tumor accumulation Massive tumor cell apoptosis	[76]
NO-responsive NP	NO donor	SMA-tDodSNO and SMA		DOX	-	-	4T1 cells bearing mice	A synergistic effect on cell survival with an IC_50_ of 1.79 ± 0.7 nM 87.4% of cell population in subG1 phase A drop in the alive cells’ percentage to 21.7 ± 3.9% Significant tumor growth inhibition	[77]
Hypoxia-responsive NP	Hypoxia	Hybrid PLGA lipid NPs (DPPC, DSPE-PEG and DSPE-PEG-DBCO)		TPZ	iRGD	αυ-integrins neuropilin-1	4T1 cells bearing mice	Efficient targeting with almost 2-fold increase in comparison with non-targeted particles Significant cell cytotoxicity (IC_50_ in hypoxia: 3.7 μg/mL, IC_50_ in normoxia: 9.4 μg/mL) Possessing highest cellular uptake in spheroids High tumor penetration Strong tumor cell killing Successful metastasis inhibition	[78]
Thermo-responsive NP	High temperature	poly ((2-(2-methoxyethoxy) ethyl methacrylate-*co*-oligo (ethylene glycol) methacrylate)-*co*-2-(dimethylamino) ethyl methacrylate-*b*-poly (d,l-lactic-coglycolide) and PDA as film coating		DOX PTX	siRNA	Survivin	MDA-MB-231 bearing mice	80% tumor cell death Sensitized cancer cells to chemotherapy Non-significant adverse effects	[72]

NP, nanoparticle; TNBC, triple negative breast cancer, ROS, reactive oxygen species; DOX, doxorubicin; TPZ, tirapazamine; PTX, paclitaxel; DTX, docetaxel; GEM, gemcitabine; NO, nitric oxide; MMP-2, matrix metalloproteinase-2; GSH, glutathione; GNP, gelatin nanoparticles; PLGA, poly(lactic-co-glycolic acid); DPPC, 1,2-dipalmitoyl-sn-glycero-3-phosphocholine; PEG, polyethylene glycol; IC_50_, half-maximal inhibitory concentration; HA, hyaluronic acid; LRP1, low density lipoprotein-receptor 1; AuNPs, gold nanoparticles; GNPs, gelatin NPs; G-AuNP, gold nanoparticles were conjugated to gelatin NPs; DGL, dendrigraft poly-lysin; PDA, polymerized dopamine; DPPC, 1,2-dipalmitoyl-sn-glycero-3-phosphocholine; DSPE, 1,2-distearoyl-sn-glycero-3-phosphoethanol-amine; DBCO, dibenzocyclooctyl; SMA, styrene-maleic acid; SMA-tDodSNO, SMA-tert-dodecane *S*-nitrosothiol; RRGD, tandem peptide of RGD (arginyl-glycyl-aspartic acid) and octaarginine; FPR-1, formyl peptide receptor 1; Combo NCs, cationic liposome nanocomplex; TEMPO, nitroxide radical (2,2,6,6-tetramethylpiperidine-1-yl)-oxyl; ADD, adjudin; Pt, cisplatin; NIR, near infra-red; RNPs, ROS scavenging nitroxide radical-containing NPs; RNP^N^, pH-sensitive RNP; RNP^0^, pH insensitive RNP.

## Data Availability

No new data were created or analyzed in this study. Data sharing is not applicable for this article.

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
