# Peer review of "Bio-Inspired and Smart Nanoparticles for Triple Negative Breast Cancer Microenvironment"

_pharmaceutics, 2021, doi:10.3390/pharmaceutics13020287_

Round 1

Reviewer 1 Report

The review seems thorough and well written, but there are a few grammatical errors throughout.

For example, the sentence starting on line 18 needs to be rewritten for grammar.

On line 30: “Among the different 30 criteria for BC classification, two features are commonly considered regarding the 31 expression or deficiency of specific receptors: human epidermal growth receptor 2 (HER2), 32 progesterone receptors (PR), and estrogen receptors (ER) [3]” - you say two features but list 3?

It would be informative to the reader to give a few sentence synopsis of how the membrane coated NPs are typically produced.

Reviewer 2 Report

TNBC is an aggressive tumor, prone to metastasize and characterized by high heterogeneity. Currently no FDA-approved targeted therapies exist. In that, TNBC treatment is a hot topic. 

This review summarizes the intensive efforts made to deliver accurate amounts of therapeutic drugs to TNBC relying on cell-membrane and well presents the state-of-the-art; in my opinion it deserves to be published.

Although the text is well organized and some explanatory figures are included, this reviewer suggests to add additional figures, for example depicting the general structure of  PLGA NP or iRGD-PLGA-ICG-TPZ or the repetitive monomer of PDA .. this would help the reader to appreciate the different nano materials reported also in terms of chemical properties and feasibility.

Some misprints shall be checked.

Reviewer 3 Report

In this review, the authors provide a detailed overview of various nanoparticles/nanosystems that can modulate/target the tumor ECM specifically for TNBC therapy. The review includes a comprehensive description of the studies included and good figures and informative table.

However, there are a few concerns that should be addressed by the authors before the review can be accepted for publications.

Major points:

1)  In general, while most sections have a straightforward and at times very detailed description of the results of the cited studies, they are missing the in-depth insight, analyses and pros/cons (from the side of the authors) of the cited studies. Adding these will enhance the quality of the review and add more depth to it.

2) Lines 106-107. There is no real transition from "existing therapies" to the "bioinspired nanosystems". While section 2 summarizes results of existing therapies, this section should build a strong case for why tumor-homing systems would be required and/or introduced in section 3. This is missing in section 2, which currently seems to imply that chemo works. e.g. chemotherapy like platinum is actually better than taxane for BRCA1 mutations. In this case, there would seem to be no need for tumor-homing systems.

3) English: It is strongly recommended that the review be proofread and edited by a native English speaker to improve grammar, consistency in present/past tense usage and some sentence construction. For example, line 19: "is an urgent need" should be "are an urgent need", line 59: were all identified: switched from present tense to past tense, line 65: "which all require for an urgent need" should be "which all require an urgent need", line 109: enhanced is repeated twice, lines 114-116 are almost similar as lines 125-127. line 134: "adoptable" shud be "adaptable", etc.

4) Fibroblasts form a critical part of the tumor ECM and it will help to include some points about studies targeting cancer-associated fibroblasts.

Other minor points:

5) Lines 18-20: Do you mean "patients who do respond to" or "patients who do not respond to" current chemo?

6) Line 37: suggest replacing "thick" with "dense"

7) Lines 108-114 should have multiple citations for the relevant systems, toxicities, etc.

8) Suggest increasing font size/resolution in Figure 4 and 5 for clarity.
